# Diagnostic of fatty liver using radiomics and deep learning models on non-contrast abdominal CT

**Haoran Zhang**[1], **Jinlong Liu**[1], **Danyang Su**[1], **Zhen Bai**[1,2], **Yan Wu**[1,2], **Yuanbo Ma**[1], **Qiuju Miao**[1,2], **Mingyue Wang**[1], **Xiaopeng Yang**[1,2]*

1 Department of Radiology, The First Affiliated Hospital of Zhengzhou University, Zhengzhou, Henan Province, China, 2 Department of Medical Equipment, The First Affiliated Hospital of Zhengzhou University, Zhengzhou, Henan Province, China

* 13837141925@163.com

**Data Availability Statement:** We have included the minimal data set used in our experiments within the supplementary materials, which encompasses clinical data, detailed 2D and 3D radiomics features

## Abstract

### Purpose

This study aims to explore the potential of non-contrast abdominal CT radiomics and deep learning models in accurately diagnosing fatty liver.

### Materials and methods

The study retrospectively enrolled 840 individuals who underwent non-contrast abdominal CT and quantitative CT (QCT) examinations at the First Affiliated Hospital of Zhengzhou University from July 2022 to May 2023. Subsequently, these participants were divided into a training set (n = 539) and a testing set (n = 301) in a 9:5 ratio. The liver fat content measured by experienced radiologists using QCT technology served as the reference standard. The liver images from the non-contrast abdominal CT scans were then segmented as regions of interest (ROI) from which radiomics features were extracted. Two-dimensional (2D) and three-dimensional (3D) radiomics models, as well as 2D and 3D deep learning models, were developed, and machine learning models based on clinical data were constructed for the four-category diagnosis of fatty liver. The characteristic curves for each model were plotted, and area under the receiver operating characteristic curve (AUC) were calculated to assess their efficacy in the classification and diagnosis of fatty liver.

### Results

A total of 840 participants were included (mean age 49.1 years ± 11.5 years [SD]; 581 males), of whom 610 (73%) had fatty liver. Among the patients with fatty liver, there were 302 with mild fatty liver (CT fat fraction of 5%–14%), 155 with moderate fatty liver (CT fat fraction of 14%–28%), and 153 with severe fatty liver (CT fat fraction >28%). Among all models used for diagnosing fatty liver, the 2D radiomics model based on the random forest algorithm achieved the highest AUC (0.973), while the 2D radiomics model based on the Bagging decision tree algorithm showed the highest sensitivity (0.873), specificity (0.939), accuracy (0.864), precision (0.880), and F1 score (0.876).

for model construction, labels and group information used in model building, and liver fat content measured by QCT, among other data. These data included in the supplementary materials will facilitate the provision of information for the model construction described in the paper and meet the requirements for reproducibility and validation. The non-author contact for the datasets used in our study was Jiahe Zhang, who is a member of the Data Access Committee of the First Affiliated Hospital of Zhengzhou University. The contact information is as follows: Email: 3212759684@qq.com.

**Funding:** This study was financially supported by the Medical Science and Technology Research Program of Henan Province in 2022 in the form of a grant (SBGJ202102089) received by XY. No additional external funding was received for this study. The funder had no role in study design, data collection and analysis, decision to publish, or preparation of the manuscript.

**Competing interests:** The authors have declared that no competing interests exist.

## Conclusion

A systematic comparison was conducted on the performance of 2D and 3D radiomics models, as well as deep learning models, in the diagnosis of four-category fatty liver. This comprehensive model comparison provides a broader perspective for determining the optimal model for liver fat diagnosis. It was found that the 2D radiomics models based on the random forest and Bagging decision tree algorithms show high consistency with the QCT-based classification diagnosis of fatty liver used by experienced radiologists.

## Introduction

Hepatic steatosis is characterized by the accumulation of lipids within hepatocytes and is a hallmark of non-alcoholic fatty liver disease (NAFLD) [1]. With the increasing incidence of obesity, NAFLD affects approximately 25.24% of adults worldwide, making it the most prevalent liver disease [2–5]. Due to its high prevalence, NAFLD has become the fastest-growing cause of liver-related mortality globally and is increasingly recognized as a significant contributor to end-stage liver disease, indications for liver transplantation, and hepatocellular carcinoma [6–9]. Consequently, this poses a substantial burden on healthcare resources. Early screening and diagnosis of fatty liver are crucial for timely intervention in the condition of patients with fatty liver, delaying or preventing further progression of the disease.

Currently, in clinical practice, liver biopsy is the gold standard for diagnosing NAFLD. However, the routine implementation of this screening method is hindered due to its invasive nature, potential risks, and associated complications such as internal bleeding, bile leakage, and infection. In recent years, with the continuous advancement of imaging technology, it has played an increasingly important role in the non-invasive assessment of fatty liver. Traditional ultrasound (US) is widely used for fatty liver screening due to its convenience and cost-effectiveness. However, it has subjectivity in the diagnostic process and relatively low diagnostic sensitivity for mild fatty liver disease [10]. Proton density fat fraction (PDFF) obtained through chemical shift-encoded magnetic resonance imaging has been established as a non-invasive imaging biomarker for assessing hepatic steatosis [11–13]. However, this method is costly and requires subjects to hold their breath for extended periods [14, 15]. Furthermore, MRI cannot measure PDFF in patients with claustrophobia or those with metallic implants. In contrast, CT allows for rapid image acquisition and minimizes patient discomfort [16–18]. With the increasing public concern over gastrointestinal cancer screening, the clinical use of chest and abdominal CT has significantly increased, thereby expanding its potential for diagnosing fatty liver [19–21].

It is worth noting that the rapid development of artificial intelligence (AI) technology has brought new opportunities to the field of medical diagnostics [22–27]. The seamless integration of AI with CT technology holds great potential for improving the accuracy and efficiency of fatty liver diagnosis. Previous studies have focused on specific algorithms or technologies [28]; for example, Graffy [29] et al. primarily utilized 3D convolutional neural networks within deep learning (DL) algorithms to quantify liver fat, while Yoo [30] et al. employed specialized DL algorithms for organ segmentation and assessment of related parameters. In contrast, this study employed twelve machine learning algorithms and the ResNet18 DL network, utilizing both 2D and 3D models to facilitate multidimensional analysis of liver imaging data. This comprehensive approach covers a broader range of algorithm types, enabling extensive exploration and examination of liver imaging data from different perspectives. This diversity may

reveal new and more promising diagnostic patterns while providing advanced technological tools for the diagnosis of fatty liver.

## Materials and methods

### Study cohort

Retrospectively collected individuals who underwent abdominal non-contrast CT and quantitative CT (QCT) [17, 31] scans at the First Affiliated Hospital of Zhengzhou University from July 2022 to May 2023. A single scan was sufficient for both non-contrast CT and QCT, without additional radiation exposure. After excluding subjects with the following issues, a total of 840 individuals were included: (1) algorithm failure; (2) missing data; (3) poor image quality. They were divided into a training set (n = 539) and a test set (n = 301) in a 9:5 ratio (Fig 1); the training set included 372 males and 167 females, aged 18–86, with an average age of 48.96 ±11.24 years; Based on QCT liver fat content measurements, 148 individuals were diagnosed with normal liver health, and 391 with fatty liver disease. Among them, 195 had mild fatty liver, 99 had moderate fatty liver, and 97 had severe fatty liver. The test set included 209 males and 92 females, aged 22–90, with an average age of 49.33±11.93 years. Based on QCT liver fat content measurements, 82 individuals were diagnosed with normal liver health, and 219 with fatty liver disease, including 107 with mild fatty liver, 56 with moderate fatty liver, and 56 with severe fatty liver. The study was approved by the ethics committee (approval number 2022-KY-0961-002), and the requirement for informed consent was waived.

### Equipment and parameters

The abdominal scans were performed using a Philips Healthcare scanner (Brilliance iCT Elite FHD, OH, USA), covering the area from above the diaphragm to the lower edge of the liver. Scan parameters included a tube voltage of 120 kvp, a tube current of 150–200 mAs automatically adjusted based on patient weight, a field of view of 500×500 mm, and a slice thickness of 1.25 mm. Images were reconstructed using a standard algorithm with a tube voltage of 120 kvp, a tube current of 150–200 mAs, a slice thickness and interval of 1.25 mm, and a reconstruction field of view of 400×400 mm. The CT scanner underwent daily calibration for quality control.

### Post-image processing

The images were imported into the post-processing workstation of the image analysis system (QCT Pro 6.1) to measure the average CT values of the standard phantom density material (Model 4) for quantitative liver fat analysis. Skilled radiologists manually delineated quasi-circular regions of interest (ROI) on the liver sections, specifically at the cross-section where the right branch of the portal vein enters the liver, ensuring that major blood vessels and bile ducts were avoided. ROI of uniform size were placed in the peripheral regions of the left lobe, the right anterior lobe, and the right posterior lobe, with three ROI used to measure the average liver fat content.

### Liver segmentation

A radiologist with extensive experience but blinded to the study details manually segmented the ROI in the CT images of the liver using ITK-Snap software (version 3.6.0). To enhance the continuity of features, we selected the slice with the largest cross-sectional area of the liver and the two slices immediately above and below it to create a 2D ROI for liver steatosis analysis. One month later, we randomly selected 30 cases from the training cohort to assess the reproducibility of radiomic features based on 2D ROI in a blinded manner. Features with an

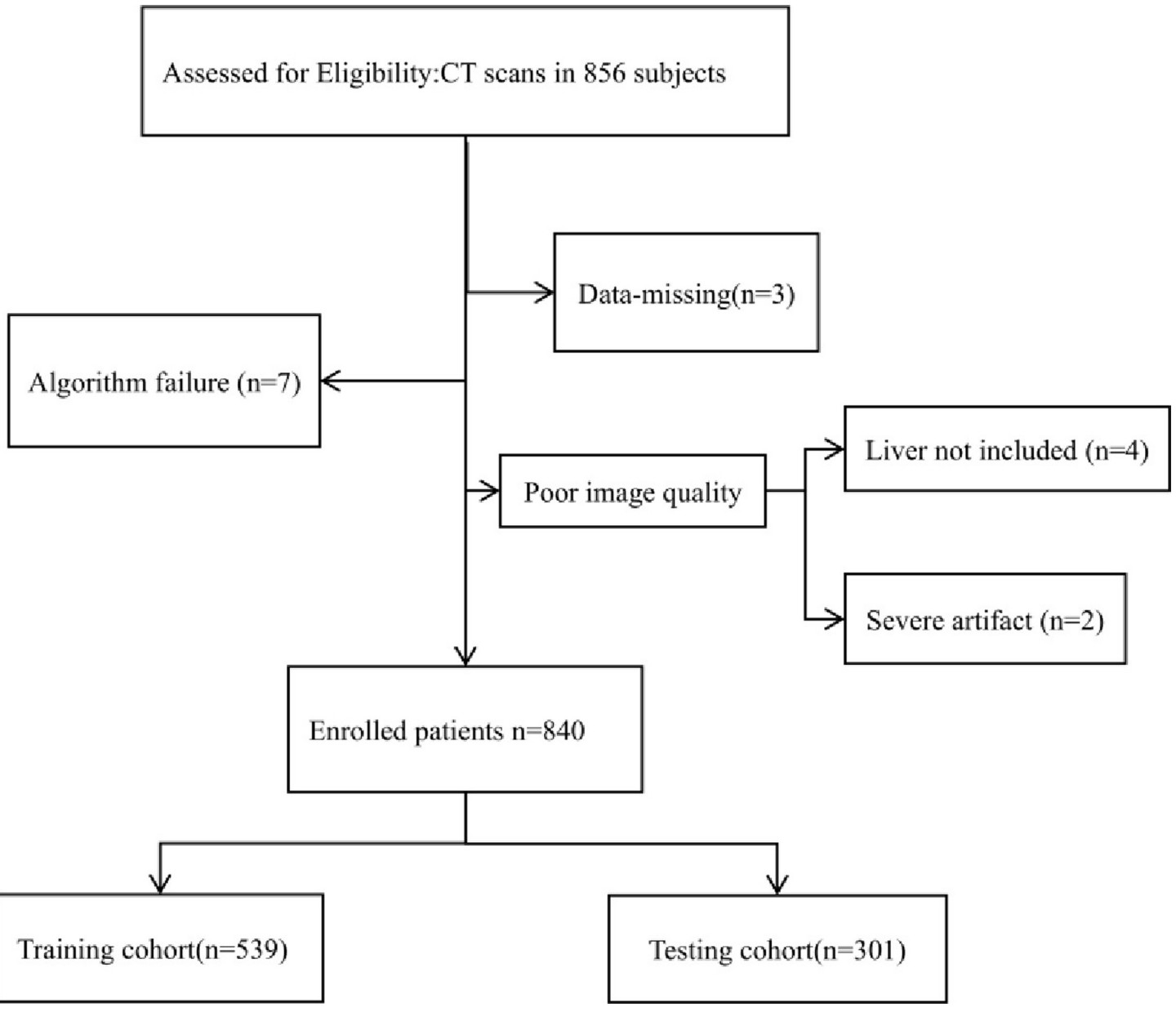

**Fig 1. Participant flow chart.**

intraclass correlation coefficient (ICC) greater than 0.75 were selected for analysis. We employed a liver segmentation model based on the Vector Boosting network (VB-Net) [32, 33] to automatically segment the entire liver. The network structure is visible in S1 Fig. Subsequently, morphological erosion was applied to process the images and extract the liver, serving as a 3D ROI for liver steatosis analysis (Fig 2).

EFECV, Recursive Feature Elimination with Cross - Validation; LASSO, the least absolute shrinkage and selection operator; AUC, area under the receiver operating characteristic curve; 0, normal liver health; 1, mild fatty liver; 2, moderate fatty liver; 3, severe fatty liver ;ROC, Receiver Operating Characteristic.

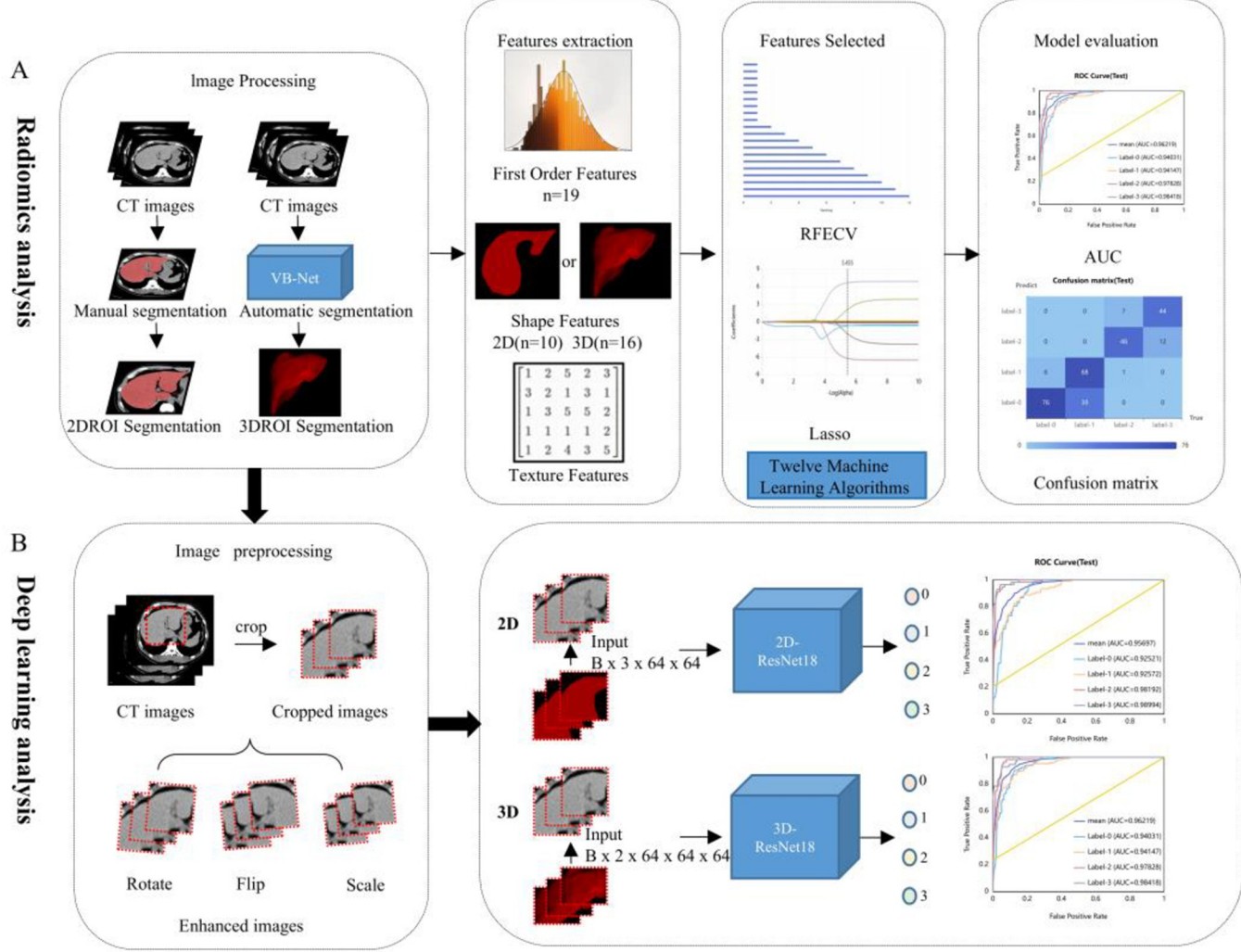

**Fig 2. Analysis flowchart.**

## Radiomics models construction

**Radiomic feature extraction.** Utilizing the Python package Pyradiomics, we automatically extracted radiomic features from radiologists' ROI. The radiomic features were standardized in accordance with the Image Biomarker Standardization Initiative (IBSI). The study was conducted following the reporting guidelines of the IBSI. A total of 2264 2D and 2264 3D liver radiomic features were extracted, encompassing first-order, second-order, and shape features [34].

**Feature selection and model construction.** After standardizing the two-dimensional and three-dimensional liver imaging features to have a mean of 0 and a standard deviation of 1, we applied a transformation function $Z = (x - \mu) / \sigma$, where Z represents the standardized score (normalized value). In the function, x and Z denote the pixel intensities of the original and normalized images, respectively, while $\mu$ and $\sigma$ represent the mean and standard deviation of the original image intensity values. Subsequently, we employed five-fold cross-validation, recursive feature elimination, and the least absolute shrinkage and selection operator (LASSO) algorithm for feature selection (Fig 2). Features were selected and input into the liver 2D radiomics model and the 3D radiomics model for training, respectively. The characterisation of the

model is visible in the S1 and S2 Tables. Validation of features visible S3 Table. Liver fat classification models were established using algorithms such as adaptive boosting (AdaBoost), gradient boosting decision tree (GBDT), K-nearest neighbors (KNN), eXtreme gradient boosting (XGBOOST), decision trees (DT), gaussian processes (GP), bagging decision trees (BDT), logistic regression (LR), stochastic gradient descent (SGD), support vector machines (SVM), quadratic discriminant analysis (QDA), and random forests (RF).

## Clinical model construction

The clinical model was developed using the clinical features listed in Table 1. These features were standardized using the z-score normalization method. Subsequently, LASSO regression was used for clinical feature selection, followed by training classifiers for the prediction of liver steatosis classification. Full details can be found in the S4 Table.

## DL model construction

**Data preprocessing.** Initially, a relatively balanced subset of data is obtained through balanced sampling. The balanced samples are then processed using the FixedNormalizer normalization method. Operations are conducted with a mean of 40 and a standard deviation of 110. The distribution of the data is adjusted to a more suitable form. Concurrently, the clip operation is employed to prevent potential extreme values in the data from interfering with subsequent model training. The sampling method for the 3D DL model is of fixed length, with a spacing of [3.0, 3.0, 3.0], a random box center at [5, 5, 5], and a cropping size of [64×64×64]. The sampling method for the 2D DL model is of fixed length, with a spacing of [3.0, 3.0], a random box center at [5, 5], and a cropping size of [64×64].

**Data enhancement.** In DL model training, data augmentation is employed to increase the diversity and quantity of training data, thereby enhancing the model's generalization capabilities. Data augmentation in this study is achieved through various methods, including random

Table 1. Baseline characteristics of clinical and examination results.

| Variables | Training cohort | Test cohort | p value |
|---|---|---|---|
| Number of participants | 539 | 301 | / |
| Age (years) | 50(41,57) | 49(42,57) | 0.963 |
| Gender (male) | 372(69.0%) | 209(69%) | 0.900 |
| Low-Density Lipoprotein | 2.84±0.85 | 2.94±0.83 | 0.116 |
| Total Cholesterol | 4.76(4.12,5.45) | 4.84(4.17,5.54) | 0.324 |
| Glucose | 5.31(4.94,5.92) | 5.33(4.97,5.89) | 0.903 |
| Alanine Aminotransferase | 24(16,39) | 23(15,43.5) | 0.846 |
| Aspartate Aminotransferase | 22(18,28) | 22(17,30) | 0.901 |
| Glutamyl Transpeptidase | 27(17,45) | 27(17,46.5) | 0.743 |
| Alkaline Phosphatase | 75(62,88) | 76(63,88.5) | 0.399 |
| Total Protein | 72.31±4.26 | 72.39±4.48 | 0.800 |
| Albumin | 47.3(45.7,49.3) | 47.8(45.7,49.6) | 0.329 |
| Total Bilirubin | 10.71(8.1,13.88) | 10.8(8.29,13.72) | 0.610 |
| Direct Bilirubin | 4.35(3.5,5.51) | 4.4(3.46,5.54) | 0.823 |
| Indirect Bilirubin | 6.4(4.5,8.6) | 6.3(4.65,8.35) | 0.686 |
| Triglycerides | 1.58(1.1,2.3) | 1.64(1.12,2.42) | 0.542 |
| High-Density Lipoprotein | 1.18(1.02,1.39) | 1.19(1.02,1.42) | 0.997 |
| Globulins | 24.86±3.82 | 24.76±3.42 | 0.700 |

flipping, rotation, and scaling. Random flipping encompasses both horizontal and vertical flips. This augmentation method increases data diversity, particularly in images with strong symmetry, where the effect is pronounced. Random rotation can be performed within any angle range, with a setting range of -5˚ to +5˚. Random scaling can alter the size of the image, typically scaled within a certain range, with a scaling range set from 1 to 2 times, followed by cropping or padding back to the original dimensions.

**Model construction.**   The 2D ResNet18 and 3D ResNet18 network models were utilized, training 8 samples at a time for 500 iterations with 4 IO threads. The learning rate update method was set to step size 1000, gamma 0.1, and last epoch -1, with an initial learning rate of 0.0001. The Focal loss function was employed with target weights set to 0.25, 0.25, 0.25, 0.25, and a loss focal gamma of 2. The Adam optimizer was used with Betas of 0.9 and 0.99.

## Statistical analysis

Categorical data are assessed using chi-square tests or Kappa tests, while continuous data are analyzed using Mann-Whitney U tests or independent t-tests. The performance of the predictive model is evaluated by analyzing the receiver operating characteristic (ROC) curve and calculating key metrics such as area under the receiver operating characteristic curve (AUC), accuracy, sensitivity, and specificity. To establish the 95% confidence interval for AUC, the cl. auc function from the pROC package in R was utilized. SPSS software version 27.0 and R software version 3.5.1 were employed to facilitate the execution of all statistical analyses.

## Results

### Clinical characteristics

The clinical characteristics of the two cohorts are presented in Table 1. There were no statistically significant differences between the training and testing sets in terms of gender, age, low-density lipoprotein, and total cholesterol.

### Radiomics model diagnostic performance

In the 2D radiomics models, the model utilizing the RF classifier achieved the highest AUC (0.973) for liver steatosis classification (Table 2). The model based on the BDT classifier demonstrated the highest sensitivity (0.873), specificity (0.939), accuracy (0.864), precision (0.880),

**Table 2. Performance of 2D radiomics models with different classifiers.**

| Method(2D) | AUC(95% CI) | sensitivity | specificity | accuracy | precision | f1Score |
|---|---|---|---|---|---|---|
| AdaBoost | 0.968(0.950–0.985) | 0.850 | 0.929 | 0.841 | 0.855 | 0.852 |
| BDT | 0.967(0.947–0.984) | 0.873 | 0.939 | 0.864 | 0.880 | 0.876 |
| DT | 0.964(0.948–0.980) | 0.846 | 0.926 | 0.841 | 0.861 | 0.851 |
| GP | 0.961(0.944–0.978) | 0.850 | 0.930 | 0.841 | 0.855 | 0.852 |
| GBDT | 0.962(0.941–0.981) | 0.843 | 0.929 | 0.834 | 0.846 | 0.845 |
| KNN | 0.959(0.936–0.979) | 0.847 | 0.928 | 0.837 | 0.852 | 0.849 |
| LR | 0.927(0.896–0.957) | 0.747 | 0.894 | 0.741 | 0.760 | 0.751 |
| QDA | 0.957(0.937–0.975) | 0.792 | 0.907 | 0.787 | 0.804 | 0.797 |
| RF | 0.973(0.959–0.986) | 0.871 | 0.936 | 0.860 | 0.877 | 0.873 |
| SGD | 0.916(0.880–0.952) | 0.757 | 0.898 | 0.744 | 0.758 | 0.756 |
| SVM | 0.958(0.939–0.976) | 0.828 | 0.915 | 0.811 | 0.829 | 0.829 |
| XGBOOST | 0.969(0.949–0.986) | 0.847 | 0.930 | 0.837 | 0.847 | 0.847 |

and F1 score (0.876). In the 3D radiomics models (Table 3), the models employing the BDT and SVM classifiers achieved the highest AUC (0.964) for the classification of fatty liver. The model based on the BDT classifier had the highest sensitivity (0.856), accuracy (0.841), precision (0.854), and F1 score (0.854). The model based on the XGBOOST classifier exhibited the highest specificity (0.936) and accuracy (0.841). The performance of all radiomics models in classifying liver steatosis was superior to that of the model based on clinical data (AUC 0.788).

## 2D radiomics features

In the radiomics model for the classification diagnosis of fatty liver, the highest AUC, sensitivity, specificity, accuracy, precision, and F1 score were all observed in the 2D radiomics model. The features used in the 2D radiomics model were constructed from the 15 significant features remaining after applying five-fold cross-validation, recursive feature elimination, and LASSO selection from a library of 2264 2D features. The weights of the features are depicted in the Fig 3. Box plots illustrating the contribution of each feature to the classification diagnosis of fatty liver are shown in the S2 Fig.

## DL model diagnostic performance

As shown in Table 4, the accuracy and AUC for the classification diagnosis of liver steatosis for the 2D and 3D DL models are 0.797 and 0.784, and 0.957 and 0.962, respectively. Therefore, it is concluded that the performance of the 3D DL model is superior to that of the 2D DL model. Both models outperform the model based on clinical data (AUC 0.788).

## Diagnostic performance analysis

The two-dimensional radiomics model based on the RF classifier demonstrated the highest diagnostic performance for fatty liver among all models, with an AUC of 0.973. Using the diagnosis of fatty liver by experienced radiologists using QCT as the benchmark, the diagnostic performance of this optimal model was analyzed (Fig 4).

Among the 42 participants misclassified by the optimal radiomics model, 14 mild fatty liver patients were categorized as asymptomatic, 20 asymptomatic individuals were misclassified as mild fatty liver patients, 1 moderate fatty liver patient was misclassified as mild, 5 severe fatty liver patients were misclassified as moderate, and 2 moderate fatty liver patients were misclassified as severe.

Table 3. Performance of 3D radiomics models with different classifiers.

| Method(3D) | AUC(95% CI) | sensitivity | specificity | accuracy | precision | f1Score |
|---|---|---|---|---|---|---|
| AdaBoost | 0.959(0.940–0.977) | 0.839 | 0.927 | 0.824 | 0.845 | 0.840 |
| BDT | 0.964(0.946–0.980) | 0.856 | 0.933 | 0.841 | 0.854 | 0.854 |
| DT | 0.931(0.898–0.963) | 0.832 | 0.921 | 0.814 | 0.829 | 0.829 |
| GP | 0.955(0.932–0.977) | 0.815 | 0.920 | 0.804 | 0.818 | 0.815 |
| GBDT | 0.953(0.928–0.977) | 0.818 | 0.917 | 0.804 | 0.819 | 0.818 |
| KNN | 0.915(0.877–0.953) | 0.807 | 0.908 | 0.787 | 0.805 | 0.806 |
| LR | 0.939(0.913–0.964) | 0.763 | 0.903 | 0.757 | 0.763 | 0.762 |
| QDA | 0.946(0.923–0.968) | 0.791 | 0.911 | 0.774 | 0.788 | 0.786 |
| RF | 0.962(0.940–0.980) | 0.827 | 0.923 | 0.811 | 0.823 | 0.824 |
| SGD | 0.819(0.766–0.871) | 0.721 | 0.877 | 0.708 | 0.712 | 0.715 |
| SVM | 0.964(0.946–0.980) | 0.839 | 0.929 | 0.824 | 0.838 | 0.837 |
| XGBOOST | 0.962(0.943–0.980) | 0.854 | 0.936 | 0.841 | 0.853 | 0.852 |

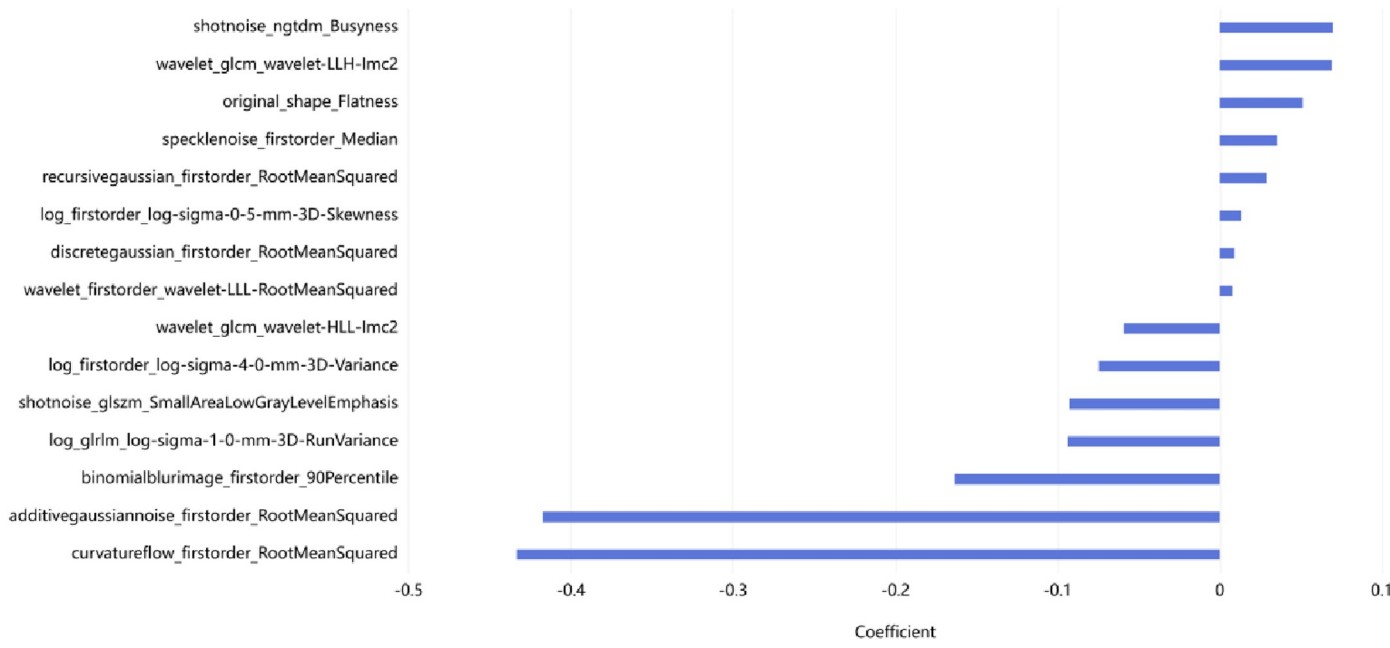

**Fig 3. Feature weighting.**

## Discussion

In this study, we conducted an in-depth investigation of the diagnostic performance of 2D and 3D radiomics models, as well as DL models, based on non-contrast abdominal CT scans in classifying four categories of liver fat. The results indicate that all models have excellent diagnostic efficacy for fatty liver, offering new perspectives and methods for the non-invasive assessment of liver fat.

In the comparison of model performance, the diagnostic efficacy of most radiomics models surpassed that of DL models. This outcome can be attributed to the unique advantages of radiomics models in processing liver imaging data. Radiomics models are capable of extracting comprehensive quantitative features from CT images, including first-order, second-order, and shape features, thereby providing a more detailed description of the texture and morphological changes in the liver. Although DL models have the potential to handle large-scale complex data, they did not fully leverage their advantages in this study, which may be due to the characteristics of the current dataset. For instance, the key features of liver fat in non-contrast abdominal CT images are already quite prominent in the 2D plane, and by focusing on these key features, radiomics models can achieve better performance. In terms of feature extraction, the 2D model effectively reduces the interference caused by redundant information that may exist in 3D data and accurately captures key features related to liver fat. This is achieved by radiologists selecting the largest cross-section of the liver and adjacent CT images for analysis

**Table 4. Comparative performance analysis of 2D and 3D DL models.**

| Method | AUC | F1score | accuracy | precision |
|---|---|---|---|---|
| 2DDLS | 0.957 | 0.807 | 0.817 | 0.797 |
| 3DDLS | 0.962 | 0.792 | 0.808 | 0.784 |

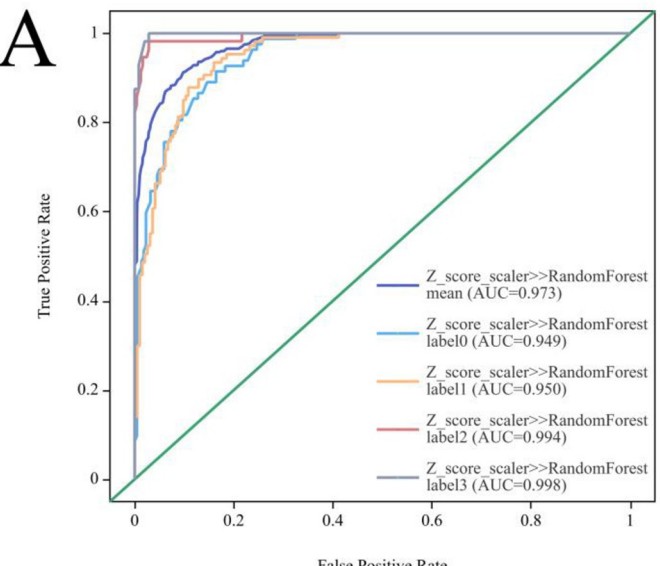

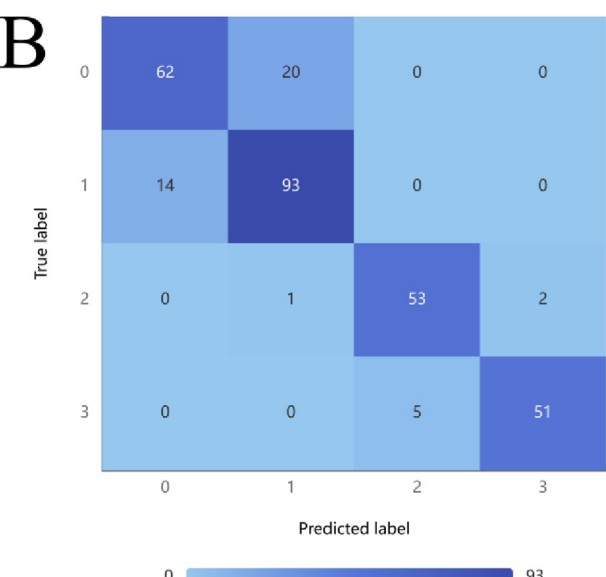

**Fig 4. Diagnostic performance.** A and B illustrate the ROC curve and confusion matrix for the 2D radiomics model, respectively. 0, normal liver health; 1, mild fatty liver; 2, moderate fatty liver; 3, severe fatty liver.

[35]. Furthermore, the structure and algorithms of the 2D model are relatively simple. Its prediction method based on linear regression allows for a more intuitive understanding of the relationship between features and weights, making the detection process easier to comprehend. Therefore, this offers more convenience for clinical applications.

This study demonstrates significant advantages over other studies related to fatty liver in several key aspects [36–38]. In terms of model comparison, this study not only explores the

differences between 2D and 3D models within the same study but also carefully analyzes the performance advantages and disadvantages of different model types. For example, compared to the 3D DL model, the 2D radiomics optimal model performs better in terms of AUC (0.973 vs. 0.962) and accuracy (0.860 vs. 0.784). In contrast, Sim [39] et al.'s study focuses solely on MR-PDFF radiomics analysis without involving DL models, Yoo [30] et al.'s study focuses exclusively on fully automated 3D organ segmentation for assessing fatty liver on CT images using a single model type, and Vianna [40] et al.'s study concentrates solely on applying DL models to grade fatty liver on ultrasound images without exploring radiomics models. In terms of study design and sample processing, this retrospective diagnostic cohort study meticulously screens and collects data from a total of 840 participants who underwent non-contrast abdominal CT examinations. The participants are randomly divided into a training set (539) and a test set (301), which enhances the generalization ability and stability of our proposed model unlike Sim et al., who had a smaller sample size of only 106 cases without mentioning subgroups or division criteria; similarly, Vianna et al.'s sample size is limited to just 199 cases with no mention of subgrouping or division methods. In terms of feature extraction and model construction, this study extracted a total of 2264 radiological features from 2D and 3D liver images using PyRadiomics, encompassing first-order, second-order, and shape features. The feature selection process involved the utilization of the z-score normalization method with recursive feature elimination strategy and LASSO regression technique. In contrast, Sim et al. only extracted 833 features from MR-PDFF images, while Vianna et al. relied on ultrasound images without such comprehensive processing. Furthermore, in this study, twelve different classifiers were trained to select the optimal model compared to Sim et al., who solely employed RF regressor for classifier diversity and comprehensiveness in model selection was lacking. When evaluating diagnostic performance, this study conducted a comprehensive analysis by combining multiple metrics including AUC, accuracy, sensitivity, and specificity; whereas Yoo et al.'s focus was primarily on CT attenuation value-related parameters without considering important metrics such as sensitivity and specificity.

From a clinical application perspective, these models offer a potentially effective tool for large-scale screening of NAFLD, thereby enhancing the early detection rate and providing more opportunities for disease prevention and treatment [41]. Compared to traditional diagnostic methods, CT image-based models enable quicker and more objective assessment of liver fat content, reducing subjectivity and human judgment errors. However, there are certain limitations in this study. The absence of secondary diagnostic confirmations such as biopsy and MRI may result in reduced sensitivity for mild cases of fatty liver, consequently affecting overall diagnostic accuracy. Furthermore, regarding model generalization, further investigation is required to assess its ability to perform well on external datasets as factors like different ethnicities, CT equipment variations, and operator differences might impact model performance. Future research can address the aforementioned limitations by increasing sample diversity, optimizing model algorithms to improve diagnostic accuracy for mild fatty liver, conducting multi-center studies to validate model generalizability, and exploring better integration of these models into clinical workflows to provide stronger support for accurate diagnosis and effective treatment of liver diseases.

## Conclusion

Focusing on non-contrast abdominal CT images, this study constructs 2D and 3D radiomics and DL models to classify liver fat into four categories for diagnosis. This research provides novel ideas and methodologies to enhance the diagnosis of fatty liver. After rigorous processing of 840 sample data and training multiple classifiers, the optimal model for 2D radiomics

demonstrates excellent performance (AUC up to 0.973, etc.) with comprehensive feature extraction that holds high clinical applicability. It is particularly suitable for opportunistic screening of NAFLD. However, there are limitations in this study such as the absence of secondary diagnostic confirmation, scope for improvement in model functionality, and the need for external validation. Overall, this study presents an effective approach for assessing liver fat; nevertheless, further enhancements are required while future studies should focus on addressing existing issues and refining diagnostic capabilities.

## Supporting information

**S1 Fig. VB-net structure.**
(DOCX)

**S2 Fig. Visualisation of 2D model features.**
(DOCX)

**S1 Table. 2D radiomics model training and test set features.**
(DOCX)

**S2 Table. 3D radiomics model training and test set features.**
(DOCX)

**S3 Table. The mean of 5 - fold cross - validation.**
(DOCX)

**S4 Table. Clinical data.** If there is still a need for explanation during the operation process, you can contact us at 13837141925@163.com.
(DOCX)

## Acknowledgments

The authors express their gratitude to the radiologists for their assistance in the diagnostic process and extend their appreciation to the head of the Radiology department for his support of this study.

## Author Contributions

**Data curation:** Jinlong Liu, Yuanbo Ma.

**Formal analysis:** Jinlong Liu, Zhen Bai.

**Investigation:** Jinlong Liu, Yan Wu, Mingyue Wang.

**Methodology:** Haoran Zhang, Qiuju Miao.

**Project administration:** Zhen Bai, Xiaopeng Yang.

**Software:** Jinlong Liu, Qiuju Miao.

**Supervision:** Danyang Su, Xiaopeng Yang.

**Validation:** Haoran Zhang.

**Visualization:** Danyang Su, Yuanbo Ma.

**Writing – original draft:** Haoran Zhang.

**Writing – review & editing:** Haoran Zhang, Xiaopeng Yang.

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
