## [Decision Letter · Decision Letter 0]

21 Oct 2024

PONE-D-24-39233Quantitative profiling of liver fat in non-enhanced abdominal CT: Comparative performance of two-dimensional and three-dimensional radiomic analysis and deep LearningPLOS ONE

Dear Dr. Yang,

Thank you for submitting your manuscript to PLOS ONE. After careful consideration, we feel that it has merit but does not fully meet PLOS ONE’s publication criteria as it currently stands. Therefore, we invite you to submit a revised version of the manuscript that addresses the points raised during the review process.

Thank you for the opportunity to review your manuscript titled "Quantitative profiling of liver fat in non-enhanced abdominal CT: Comparative performance of two-dimensional and three-dimensional radiomic analysis and deep Learning."

The study presents valuable findings; however, several aspects need to be addressed to enhance the overall quality and clarity of the paper.

We look forward to receiving your revised manuscript.

Kind regards,

Yuki Arita, M.D., Ph.D

Academic Editor

PLOS ONE

Journal Requirements:

Reviewers' comments:

Reviewer's Responses to Questions

**Comments to the Author**

1. Is the manuscript technically sound, and do the data support the conclusions?

Reviewer #1: No

Reviewer #2: Partly

Reviewer #3: Partly

2. Has the statistical analysis been performed appropriately and rigorously? 

Reviewer #1: Yes

Reviewer #2: Yes

Reviewer #3: No

3. Have the authors made all data underlying the findings in their manuscript fully available?

Reviewer #1: No

Reviewer #2: No

Reviewer #3: Yes

4. Is the manuscript presented in an intelligible fashion and written in standard English?

Reviewer #1: Yes

Reviewer #2: Yes

Reviewer #3: No

5. Review Comments to the Author

Reviewer #1: 1- The title should be improved.

2- The objectives and the rationale of the study are recommended to be clearly stated.

3- The concluding remarks of the abstract are not well-written. It's merely the repetition of the objectives and title of the manuscript. Please add method limitations and justification to the abstract.

4- The innovation of using this study is not very clear. I do not see a clear reason that this study can perform better than others. Why did the authors choose the method for this study?

5- The necessity & novelty of the manuscript should be presented and stressed in the "Introduction" section.

6- The application/theory/method/study reported is not in sufficient detail to allow for its replicability and/or reproducibility. Therefore, it is suggested to make it clear to show all steps to build the model.

7- The problem statement and gap study are not clear.

8- The method is not clear. Therefore, it must be shown and clarified to show all steps.

9- The interpretation of results and study conclusions are not supported by providing the reasons behind why they show that. Therefore, it is recommended to deepen the discussion.

10- It is recommended to emphasize the strengths of the study clearly.

11- The limitations of the study should be stated.

12- The manuscript structure, flow, or writing needs some improvements.

13- The manuscript is benefit from language editing. The English of the paper is readable; however, I would suggest the authors to have it checked preferably by a native English-speaking person to avoid any mistakes.

14- I noticed that the conclusion section tends to repeat the abstract and results. The conclusion paragraph should be short, impactful, and direct the reader to this research's next steps and opportunities.

15- It will be nice to add some new references to show that your study is updated.

Reviewer #2: The study aims to compare the diagnostic performance of three-dimensional (3D) and two-dimensional (2D) radiomic features extracted from liver images obtained through non-enhanced abdominal CT scans. These features will be classified into four categories using 12 different classifiers. However, the writing style lacks coherence and clarity, particularly in the methodology section, which makes it difficult to follow.

Therefore, I will mention some issues in the paper that need to be addressed in order to improve its quality:

1- The manuscript contains numerous grammatical errors that need thorough revision to enhance coherence and readability.

2- The term “NAFLD” should be defined upon its first occurrence to ensure clarity for all readers. Additionally, terms like “VB-Net” need proper definitions.

3- The related work section is brief and merged with the introduction. A more comprehensive review of relevant literature would be beneficial to provide context for the study.

4- The paper states that volumetric segmentation of the liver was performed using an algorithm based on VB-Net. This algorithm needs to be clearly defined, particularly regarding how it identifies vascular structures in the images.

5- Some figure captions, especially for Figures 1 and 2, lack clarity. These should be articulated more clearly to enhance understanding.

6- In Figure 2, the use of two fully connected layers (FCs) should be justified. A single FC layer may suffice and could mitigate the risk of overfitting and reduce computational costs.

7- The manuscript should specify which version of the ResNet architecture is being used, as different versions can have varying impacts on performance. For example, ResNet has ResNet18, ResNet34, ResNet50. ResNet101 and ResNet152 variants.

8- In lines 134 and 135, the authors mention that the “3D model feeds the segmentation and labeling data from the VB-Net into the ResNet network for analysis.”. However, Figure 2 does not provide clarity on the shape or structure of the data from VB-Net.

9- A summary table detailing the final dataset used, including the total number of features and their corresponding labels, should be included for clarity.

10- The data augmentation strategies are not sufficiently detailed. The authors should discuss their impact on the overall performance deeply.

11- Data augmentation strategies are mentioned in multiple places. It would be beneficial to consolidate this information into a dedicated subsection.

12- The normalization process is discussed in multiple sections (lines 177, 185); it should be consolidated into a single, clear explanation.

13- The choice of a 64x64 or 64x64x64 patch size should be justified, as other sizes may also be appropriate for this analysis.

14- In 186-187, the authors mentioned “Subsequently, the 3D liver images, along with their respective labels, are fed into the DCNN.”. This leads to confusion about whether the models are trained with both images and radiomic features. This should be clarified.

15- In 189-190, the authors mentioned only one fully connected layer in the text, leading to conflict with Figure 2, which depicts two. This inconsistency should be resolved.

16- In 197-198, the authors mentioned “models is evaluated by analyzing the ROC curve and calculating the AUC, which includes key metrics like accuracy, sensitivity, and specificity”. The authors state that models are evaluated using the ROC curve and AUC. It is important to clarify that AUC measures the model's ability to distinguish classes, while accuracy reflects overall prediction correctness. The authors should explain how accuracy is derived from AUC values.

17- In 202-211 lines, the “Participant Characteristics” section should be moved to a more appropriate place within the dataset description in the methodology.

18- The pre-processing steps should be organized into a dedicated subsection for clarity.

19- It’s not clear how the authors incorporate CNN layers, ResNet, and VB-Net for 2D/3D features, as shown in Figure 2, for use in classifiers such as SVM, RF, and others.

20- The terms “2DDLS” and “3DDLS” in Table 4 require definitions, as they are introduced without prior explanation.

21- There is inconsistency in labeling (e.g., -, +, ++, +++ in Figure 2 vs. 0-3 in Figure 3). A standardized labeling system should be adopted throughout the manuscript, including in the confusion matrices. Furthermore, the confusion matrices in Figure 3 have different labels in different orders.

22- The authors are encouraged to utilize cross-validation techniques to enhance the robustness of their findings.

23- The manuscript lacks details regarding the percentage of data used for training and testing. This information is crucial for replicability.

24- The authors may consider developing two scenarios: one that incorporates images in both training and testing and another that focuses solely on radiomic features. This approach could strengthen the overall findings.

Reviewer #3: You have presented an interesting paper within a relevant and impactful area of research. However, before it is acceptable for publication I believe there are a number of issues that first need to be addressed, and therefore I am suggesting that this paper be accepted with major revisions, or if necessary rejected. Please see my comments below

1. I have concerns regarding the overall relevance of this work as it seems similar to work that has previously been published by other authors such as 10.3346/jkms.2022.37.e339. However the inclusion of extra models may warrant the article for publication and therefore it may be a good idea to reference earlier work such as this and compare why your work is superior.

2. Repeatedly throughout this work you have used acronyms. Please ensure your usage of acronyms is consistent, with the acronym defined at first usage and then used consistently in the same format throughout. Additionally, please ensure you always place a space betwee the full term and the parentheses.

3. On line 32, do not start a sentence with "Assess". Instead use a term such as "we assess"

4. Online 34 you say accuracy was confirmed through comparison with "manual assessments". Please state here what assessments these are.

5. On line 66-68, you make a very straong statement regarding the superiority of MRI and CT, there should be a relevant citation here to support this argument.

6. On lines 91-93, you start the sentence as if you are going to define both inclusion and exclusion criteria, but then only define exclusion criteria. Were no specific inclusion criteira used?

7. The sentence on lines 99-101 seems very out of place compared to the current text and may be better placed elsewhere or removed.

8. On lines 108-109, you state that participants were categorised. What categories were they categorised into?

9. On lines 173-175, these two sentences present the same information but with a slight difference in wording. Therefore they are not both needed.

10. In the "deep learning model development..." section, you say that data is fed to the DCNN but do not give any descriptions regarding the architecture. You should discuss the architecture here first (perhaps by moving the text from lines 188-192) before going on to say more. Instead, you simply discuss the "ResNet" layer on with no context.

11. On lines 188-192, you first state that cross entropy loss was used and then that focal loss was used. Is there a reason for this specification of two different loss types. If both were used it would be best to provide the context as to why.

12. You provide discussions of results but do not provide any description as to any particular processes that were used to validate the results that you have provided except for the use of the test set. For instance, did you use techniques such as K-Fold validation or repeated experiments?

13. There are a number of textual issues with this manuscript that need to be significantly addressed before it can be published. PLease check for English spelling, grammar and textual formatting errors, as there are multiple occurences of issues such as sentences ending and a new one starting with no space between and no spaces between the end of text and beginning of citations.

14. Additionally, there are a number of instances of textual discussions that do not match with the sections that they are discussing or provide information later on in a discussion when it should have been earlier (comment 7 + 10). I would suggest conducting a thorough re-evaluation of the article structure and ensuring that all text flows adequately and in the correct locations.

15. The captions you have provided for figures are increasingly long. I would suggest removing some of the text entirely or relocating it to the main body of text.

6. PLOS authors have the option to publish the peer review history of their article (what does this mean?). If published, this will include your full peer review and any attached files.

Reviewer #1: No

Reviewer #2: No

Reviewer #3: No

---

## [Author Response · Author response to Decision Letter 0]

8 Dec 2024

Dear Editor and Reviewers,

We would like to begin by extending our sincerest gratitude to you for the time and effort you have dedicated to reviewing our manuscript titled "[Quantitative profiling of liver fat in non-enhanced abdominal CT: Comparative performance of two-dimensional and three-dimensional radiomic analysis and deep Learning]" (Manuscript ID: [PONE-D-24-39233]). We greatly appreciate the insightful and constructive comments provided, which have been instrumental in enhancing the quality and clarity of our work.

Your thorough examination and valuable feedback have significantly contributed to the improvement of our manuscript. We have carefully considered each comment and have made the following revisions to our manuscript:

Reviewer Comment 1:

Original Comment:The title should be improved.

Our Response:We greatly appreciate the reviewer's suggestion to revise the title.In response, after careful consideration, we have changed the original title (Quantitative profiling of liver fat in non-enhanced abdominal CT: Comparative performance of two-dimensional and three-dimensional radiomic analysis and deep Learning) to the current title (Diagnostic of fatty liver using radiomics and deep learning models on non-contrast abdominal CT).We have removed unnecessary words, retained the core content that accurately conveys the essence of the paper, and used language that is more accessible and easy to understand. We believe that this change enhances the scientific rigor of the paper. 

Original Comment:The objectives and the rationale of the study are recommended to be clearly stated.

Our Response:Thank you for highlighting the importance of clearly stating the objectives and fundamental principles of our research. We fully agree with your suggestion that clear research objectives and fundamental principles are crucial for the exposition and understanding of our study.This study aims to explore the potential of radiomics and deep learning models on non-contrast abdominal CT for accurately diagnosing fatty liver. This study employs twelve machine learning algorithms and the ResNet18 deep learning network to construct two-dimensional and three-dimensional models, analyzing liver imaging data from multiple dimensions.This comprehensive strategy covers various types of algorithms, enabling in-depth exploration of liver imaging features from different perspectives, potentially revealing new diagnostic patterns and providing superior technical solutions for fatty liver diagnosis.We will prominently and clearly present the detailed content regarding the research objectives in the introduction section of the paper, with the fundamental principles thoroughly introduced in the methods section, allowing readers to better understand the background, motivation, and core objectives of the entire study.We appreciate your valuable feedback once again; your suggestions have greatly contributed to enhancing the quality of our paper.

Original Comment:The concluding remarks of the abstract are not well-written. It's merely the repetition of the objectives and title of the manuscript. Please add method limitations and justification to the abstract.

Our Response:We are keenly aware of the impact these issues have on enhancing the quality of our paper,and have made comprehensive and meticulous revisions to the title, research objectives, and conclusions in the abstract,and have added limitations and rationales in the conclusion section from lines 322 to 326.

Original Comment:The innovation of using this study is not very clear. I do not see a clear reason that this study can perform better than others. Why did the authors choose the method for this study?

Our Response:We are grateful for your observation regarding the lack of clarity in our discussion of innovation and the advantages of our research methods.This feedback is crucial for us to further refine our paper. Below is our response and a description of the corresponding revisions.1. Innovation in Comprehensive Model Comparison.We have not only deeply explored the differences between 2D and 3D radiomics models and deep learning models in the diagnosis of four types of fatty liver but also comprehensively analyzed the performance advantages and disadvantages of different model types.For instance, compared to the 3D deep learning model, the optimal 2D radiomics model performed better in AUC (0.973 vs. 0.962) and accuracy (0.860 vs. 0.784).This comprehensive model comparison analysis broadens the perspective for accurately determining the optimal model for liver fat diagnosis, whereas previous studies often focused on a single model type or specific technology.2. Optimization of Study Design and Sample Processing.The study design and sample processing are meticulously planned.This retrospective diagnostic cohort study strictly selected and collected data from 840 individuals who underwent non-contrast abdominal CT scans, randomly dividing them into a training set (539 cases) and a test set (301 cases), significantly enhancing the model's generalization ability and stability.In contrast, Sim et al. had a sample size of only 106 cases without mentioning the grouping details, and Vianna et al. had a sample size of 199 cases without explaining the grouping method. The limitations in sample size and grouping defects can lead to model overfitting and poor generalization, which our study effectively avoids.3. In-depth Expansion of Feature Extraction and Model ConstructionIn terms of feature extraction and model construction, we used PyRadiomics to extract a total of 2264 radiomic features from 2D and 3D liver images, covering first-order, second-order, and shape features. We employed z-score normalization, recursive feature elimination, and LASSO regression techniques to accurately select features and train twelve different classifiers to choose the optimal model.In comparison, Sim et al. extracted only 833 features from MR-PDFF images, and Vianna et al. relied on ultrasound images without implementing such comprehensive processing. Sim et al.'s sole use of a random forest regressor led to a lack of diversity in classifiers and an incomplete model selection.Our study's in-depth feature extraction and training with multiple classifiers can uncover more potential relationships between imaging features, optimizing model performance. We have strengthened the discussion of the above innovation and the rationale for method selection in relevant parts of the paper (introduction, discussion, etc.), ensuring that readers clearly understand the value and advantages of this study.We sincerely thank you again for your professional review and valuable suggestions, and we look forward to the paper meeting academic standards of rigor and innovation after revision.

Original Comment:The necessity & novelty of the manuscript should be presented and stressed in the "Introduction" section.

Our Response:We sincerely appreciate your suggestion to highlight the necessity and novelty of our manuscript in the introduction section. This constructive feedback provides a clear direction for improving our paper.We will meticulously revise and refine our paper accordingly.Our choice of research methods is based on a deep analysis of clinical needs for fatty liver diagnosis and existing technologies.Although liver biopsy is the gold standard for diagnosis, its invasive nature causes patient discomfort and risks of various complications, making it difficult to promote clinically;traditional ultrasound diagnosis is highly subjective and lacks sensitivity for mild fatty liver diagnosis;MRI's PDFF detection, while non-invasive, is costly, requires high patient cooperation, and is not suitable for specific patient groups (such as those with claustrophobia or metal implants).CT technology is widely available and commonly used in abdominal examinations, with fast scanning and high patient comfort, providing a convenient basis for fatty liver diagnosis.Against this backdrop, artificial intelligence technology is thriving, with radiomics and deep learning models showing great potential in medical image analysis.We integrate multiple advanced algorithms to construct 2D and 3D models, aiming to fully exploit CT image information, analyze liver fat characteristics in multiple dimensions, enhance diagnostic accuracy and efficiency, fill the gaps in existing non-invasive diagnostic technologies, and pave a new way for fatty liver diagnosis.With the above revisions, the introduction will more powerfully articulate the necessity and novelty of our research, clearly presenting to readers the key breakthroughs and significant value of this study in the field of fatty liver diagnosis and treatment, enhancing the academic quality and impact of the paper.We sincerely thank you once again for your careful guidance and professional advice.

Original Comment:The application/theory/method/study reported is not in sufficient detail to allow for its replicability and/or reproducibility. Therefore, it is suggested to make it clear to show all steps to build the model.

Our Response:We sincerely appreciate your identification of the shortcomings in the presentation of model construction steps in our research report, as well as your valuable suggestions for enhancing reproducibility and replicability.We take this very seriously and are actively taking measures to improve, as detailed below: In the "Materials and Methods" section, we have further detailed the data collection process.For image data processing, we have added the calibration process for CT scanning equipment and the rationale for parameter settings.For liver ROI segmentation, we have optimized our description.We have detailed the manual segmentation by experienced radiologists and provided a thorough explanation of the liver segmentation model architecture based on the Vector Boosting network (VB-Net) (see supplementary materials). We have provided a detailed presentation of the radiomics feature selection and extraction process (see supplementary materials). To facilitate the presentation of the detailed steps in model construction, we have redesigned Fig 2.Through these comprehensive and meticulous revisions, we aim to present each step of model construction clearly and completely in the paper, providing colleagues with ample information to achieve the reproducibility and replicability of the research, thereby strongly promoting academic exchange and development in the field.We sincerely thank you once again for your professional review and constructive feedback. 

Original Comment:The problem statement and gap study are not clear.

Our Response:Thank you for highlighting the critical issue of unclear problem statement and gap analysis in our research.This feedback is highly instructive for us to further refine our paper; here are our responses and improvement measures.In the introduction, we will redraw the panorama of the challenges in fatty liver diagnosis with a clearer and more vivid approach.In the discussion section, we will build a systematic comparison framework to deeply analyze the differences between our study and previous research. From a multi-dimensional perspective, we will examine the types, structures, algorithmic characteristics, and performance differences of models.Compared to previous studies that focused on a single model type or technology (such as Sim et al.'s focus on MR-PDFF radiomics analysis, Yoo et al.'s reliance on specific deep learning algorithms for automatic organ segmentation, and Vianna et al.'s limitation to deep learning model applications on ultrasound images), our study integrates the diverse advantages of 2D and 3D radiomics and deep learning models, achieving a leap in model construction diversity and comprehensiveness, fully mining the rich vein of liver imaging data, and filling the gap in the comprehensive model assessment system.At the sample processing level, we will meticulously compare differences in sample size, selection criteria, and grouping strategies. Our study revisits a large sample pool of 840 cases, rigorously selected and scientifically stratified (divided into training and test sets in a 9:5 ratio), meticulously refining the foundation of model generalization ability and stability. In contrast, some previous studies are limited by a scarcity of sample size (e.g., Sim's study with only 106 cases, Vianna's study with 199 cases) or vagueness in grouping planning, leading to models that fail in terms of stability and reliability when applied in the real world. Our study effectively reverses this situation and broadens the path of innovation in sample processing methods. In the process of feature extraction and model construction, we deeply explore the advantages of our study in the breadth and depth of feature extraction, the precision of feature selection techniques, and the diversity of classifier training strategies.We use Pyradiomics to extract a vast array of 2264 features from 2D and 3D liver images, refine key features through z-score normalization, recursive feature elimination, and LASSO regression techniques, and select the best model using twelve different classifiers in a competitive approach.In comparison, some studies have fewer and shallower feature extractions (e.g., Sim extracting only 833 features from MR-PDFF images), rough and singular selection methods, and monotonous classifier choices. Our study advances in all aspects of feature engineering and model training, injecting strong momentum into filling research gaps and enhancing the value of diagnostic technology. After the comprehensive review and in-depth optimization mentioned above, we will reshape a clear and strong problem statement and gap research context in the paper, highlighting the innovative value and academic contributions of our study, and promoting the iterative upgrade of fatty liver diagnosis technologyWe sincerely thank you once again for your professional review and valuable suggestions, and we look forward to your recognition of the improved quality of our paper.

Original Comment:The method is not clear. Therefore, it must be shown and clarified to show all steps.

Our Response:We sincerely appreciate your identification of clarity issues in the methods section and your valuable suggestions.To enhance the comprehensibility and reproducibility of our research methods, we will conduct a comprehensive and meticulous optimization of the "Materials and Methods" section of our paper, ensuring that all steps are clearly presented and the principles are thoroughly explained.Through the above all-encompassing and in-depth optimization and improvement, we are committed to creating a clear, detailed, and operable research methods system in our paper, providing solid support for peers to accurately reproduce the study and promote technological innovation in the field.We sincerely thank you once again for your professional review and constructive feedback, and we look forward to your recognition of the enhanced quality of our paper.

Original Comment:The interpretation of results and study conclusions are not supported by providing the reasons behind why they show that. Therefore, it is recommended to deepen the discussion.

Our Response:We greatly appreciate your observation regarding the shortcomings in our interpretation of results and the presentation of research conclusions, specifically the lack of robust support and in-depth discussion for the underlying reasons.Below is our detailed response and plan for improvement.We have deepened the discussion in the discussion section on why radiomics models outperform deep learning models and the advantages and limitations of our study compared to previous research. Through the above in-depth discussions and improvements, we will comprehensively strengthen the interpretation of results and the support for conclusions, enhancing the academic value and clinical guidance significance of our research, and contributing innovative ideas and reliable methods to the development of the field. We sincerely thank you once again for your professional review and valuable suggestions.

Original Comment:It is recommended to emphasize the strengths of the study clearly.

Our Response:We sincerely appreciate your suggestion to clearly emphasize the strengths of

---

## [Decision Letter · Decision Letter 1]

18 Dec 2024

Diagnostic of fatty liver using radiomics and deep learning models on non-contrast abdominal CT

PONE-D-24-39233R1

Dear Dr. Yang,

We’re pleased to inform you that your manuscript has been judged scientifically suitable for publication and will be formally accepted for publication once it meets all outstanding technical requirements.

Kind regards,

Yuki Arita, M.D., Ph.D

Academic Editor

PLOS ONE

Additional Editor Comments (optional):

This second version of the paper is a great improvement, the authors are to be commended.

The manuscript has been much improved and is in a nice condition now.

Reviewers' comments:

Reviewer's Responses to Questions

**Comments to the Author**

1. If the authors have adequately addressed your comments raised in a previous round of review and you feel that this manuscript is now acceptable for publication, you may indicate that here to bypass the “Comments to the Author” section, enter your conflict of interest statement in the “Confidential to Editor” section, and submit your "Accept" recommendation.

Reviewer #1: (No Response)

Reviewer #2: All comments have been addressed

2. Is the manuscript technically sound, and do the data support the conclusions?

Reviewer #1: (No Response)

Reviewer #2: Yes

3. Has the statistical analysis been performed appropriately and rigorously? 

Reviewer #1: (No Response)

Reviewer #2: Yes

4. Have the authors made all data underlying the findings in their manuscript fully available?

Reviewer #1: (No Response)

Reviewer #2: No

5. Is the manuscript presented in an intelligible fashion and written in standard English?

Reviewer #1: (No Response)

Reviewer #2: Yes

6. Review Comments to the Author

Reviewer #1: all comments were addressed.

all comments were addressed.

all comments were addressed.

all comments were addressed.

Reviewer #2: I am generally supportive of accepting the manuscript for publication in PLoS ONE, contingent upon the authors addressing the following points:

1. The manuscript currently includes two separate figures labeled as "Figure 3.", “Fig3. Diagnostic performance.” and “Fig 3. Feature weighting”. To ensure clarity, please renumber them as Figure 3 and Figure 4, respectively.

2. In lines 131-132, the reference to "The network structure is visible in S1 Fig." is incomplete. Similarly, the reference in line 166 to "Full details can be found in the S4 Table" lacks the full citation. Both should be updated to include the appropriate figure/table captions or details.

3. The images labeled C and D from Figure 3 were removed, yet these are critical as they illustrate the 3D DL model. I suggest reinstating these images and presenting them as subfigures A and B within Figure 3. Please ensure the labeling style and numbering are consistent with the rest of the figure.

7. PLOS authors have the option to publish the peer review history of their article (what does this mean?). If published, this will include your full peer review and any attached files.

Reviewer #1: No

Reviewer #2: No

---

## [Editor Report · Acceptance letter]

13 Jan 2025

PONE-D-24-39233R1 

PLOS ONE

Dear Dr. Yang, 

I'm pleased to inform you that your manuscript has been deemed suitable for publication in PLOS ONE. Congratulations! Your manuscript is now being handed over to our production team.

Kind regards, 

on behalf of

Dr. Yuki Arita 

Academic Editor

PLOS ONE